Corrected: Publisher correction

# Uncovering active precursors in colloidal quantum dot synthesis

Leah C. Frenette [1] & Todd D. Krauss [1,2]

Studies of the fundamental physics and chemistry of colloidal semiconductor nanocrystal quantum dots (QDs) have been central to the field for over 30 years. Although the photophysics of QDs has been intensely studied, much less is understood about the underlying chemical reaction mechanism leading to monomer formation and subsequent QD growth. Here we investigate the reaction mechanism behind CdSe QD synthesis, the most widely studied QD system. Remarkably, we find that it is not necessary for chemical precursors used in the most common synthetic methods to directly react to form QD monomers, but rather they can generate in situ the same highly reactive Cd and Se precursors that were used in some of the original II-VI QD syntheses decades ago, i.e., hydrogen chalcogenide gas and alkyl cadmium. Appreciating this surprising finding may allow for directed manipulation of these reactive intermediates, leading to more controlled syntheses with improved reproducibility.

[1] Department of Chemistry, University of Rochester, Rochester, NY 14627-0216, USA. [2] Institute of Optics, University of Rochester, Rochester, NY 14627-0216, USA. Correspondence and requests for materials should be addressed to T.D.K. (email: krauss@chem.rochester.edu)

The thermolytic decomposition of organometallic precursors to form monomers has been the synthetic method of choice for the nucleation of semiconductor nanocrystalline quantum dots (QDs) for decades[1–3]. Early synthetic methods produced high-quality QDs through the injection of the phosphine-chalcogenide precursor trioctylphosphine selenide (TOPSe) into thermally decomposing dimethylcadmium[2]. In the past 20 years, synthetic methods have advanced substantially making the colloidal QD synthesis reaction safer, and with greater control over nanoparticle size, shape, and stoichiometry. For example, the highly toxic dimethylcadmium was replaced with cadmium carboxylates[4], secondary phosphine chalcogenides allowed for control over QD surface composition[5–7], and the use of single-source precursors drove improvements in reaction yields[8–10]; all have allowed for improved size selectivity and monodispersity. Recent synthetic breakthroughs have focused on designing precursors to better program size and reaction yield[11–13].

Contrary to the advances made in developing improved QD synthesis, elucidating the specific chemical reaction pathways taking initial molecular precursors to QDs has been difficult and limited in scope. In an early example, it was shown that the reactive chalcogenide precursor in lead chalcogenide QD syntheses was not the tertiary phosphine-chalcogenide precursor, but actually a secondary phosphine-chalcogenide impurity[5]. With respect to CdSe QDs, it was proposed that QD molecular precursors (i.e., cadmium oleate and TOPSe) form a transition state given by a Lewis acid-base complex[14–16]; which forms the CdSe bond after an attack by an alkyl carboxylate[14, 16, 17]. Noteworthy products from this reaction are trioctylphosphine oxide (TOPO) and oleic anhydride[14]. Further mechanistic studies are motivated by the need to control the kinetics of the reactive precursors in solution and eliminate extraneous reactants, enhance reproducibility, and obtain better overall control over nanoparticle growth[17, 18]. In addition, new precursors and synthetic methods can be designed with controlled decomposition to yield more highly engineered QDs of desired shapes, sizes, and compositions[11].

In the following study, we take a somewhat unique approach to our mechanistic studies by investigating the thermal decomposition of each organometallic precursor separately, but importantly, under the high temperature reaction conditions that are common in CdSe QD syntheses. Specifically, we focus on the chemical reaction mechanism of the most common precursors for making CdSe QDs: the reaction of cadmium carboxylate and tertiary phosphine selenide. We hypothesize that complex reaction intermediates, where both cadmium and selenium precursors remain intact[14, 16, 19, 20], are not likely to form given the extreme temperatures necessary for CdSe QD formation thus suggesting an alternate reaction pathway. We find that at under conditions common to CdSe QD syntheses, both the tertiary phosphine-chalcogenide precursor and metal-carboxylate precursor thermally decompose to extremely reactive species: hydrogen selenide and dialkylcadmium, respectively. On the basis of this observation, we propose a mechanism for QD synthesis in which these decomposition products react to form CdSe monomer, which then further react to nucleate nanoparticles[3, 21]. Importantly, this mechanism includes the formation of the anhydride and the phosphine oxide products observed in previous mechanism studies[14, 17], and can explain the relatively low chemical yields observed in the CdSe QD synthesis reactions of this type[16].

## Results

**Thermal decomposition of tri-*n*-butylphosphine selenide.** We studied the decomposition of tri-*n*-butylphosphine selenide (TnBPSe) in the absence of a cadmium source to model the decomposition of tertiary phosphine selenides under the conditions of typical CdSe QD syntheses. High-quality cadmium chalcogenide QDs are typically synthesized at temperatures between 270 and 350 °C. Thus, tri-*n*-butylphosphine selenide (TnBPSe), octanoic acid (a shorter chain analog for oleic acid), and tetradecane were combined and heated at 250 °C. We expected significant reactivity in the phosphine selenide, as phosphine chalcogenides are likely to decompose at these temperatures given their reactivity under similar conditions[22, 23]. In addition, alkyl tertiary phosphine selenide bond

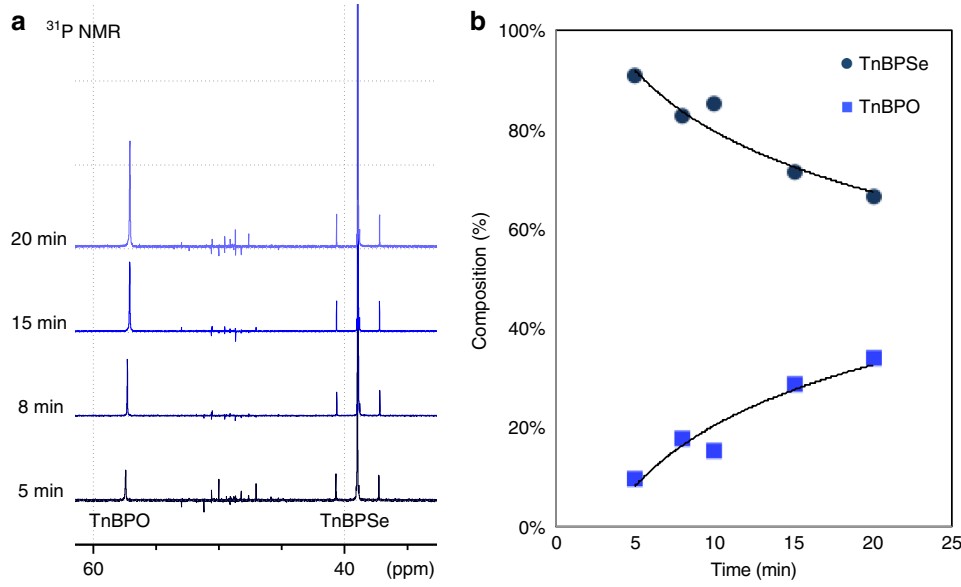

**Fig. 1** A NMR analysis following decomposition of TnBPSe after hot injection. **a** The $^{31}$P NMR spectrum of the reaction mixture over time, showing that the relative amount of TnBPSe decreases as the amount of TnBPO increases. **b** The percent composition of TnBPSe (navy blue circles) and TnBPO (royal blue squares) in the TnBPSe decomposition reaction over time. $^{31}$P and $^{77}$Se NMR of TnBPSe before reaction can be found in Supplementary Fig. 1. Trend lines are provided as guides to the eye

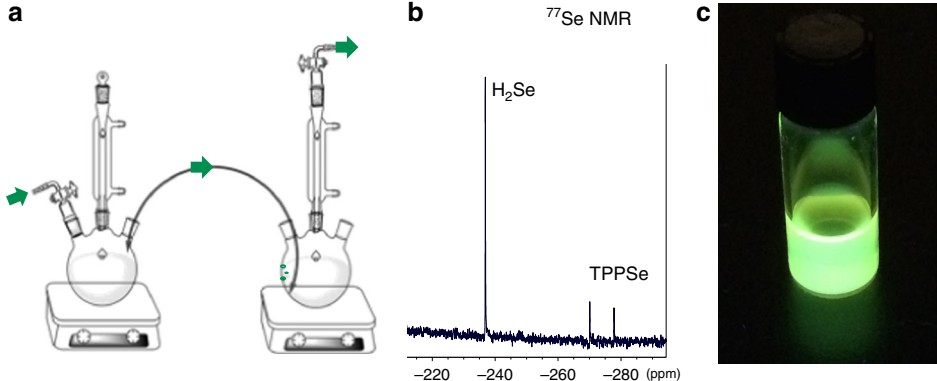

**Fig. 2** Identifying the selenium decomposition product. **a** The experimental setup used to identify H$_2$Se as the gaseous selenium source. This panel was generated using templates in ChemDraw Professional 16.0. **b** The $^{77}$Se NMR spectrum of H$_2$Se with TPPSe external standard. NMR characterization of TPPSe can be found in Supplementary Fig. 6. **c** Fluorescent CdSe nanocrystals under UV irradiation synthesized from the H$_2$Se generated in situ, the absorbance spectrum is in Supplementary Fig. 7

dissociation energies are low, making breaking the phosphine-chalcogenide bond accessible, as demonstrated by Ruberu et al. through phosphine crossover experiments at temperatures above 250 °C[24–26].

As shown in Fig. 1, analysis of the reaction mixture by $^{31}$P NMR (nuclear magnetic resonance spectroscopy) showed the formation of tri-$n$-butylphosphine oxide (TnBPO), a product universally seen in other studies of CdSe QD mechanism with cadmium present[14, 27–29], and a clear indication that the TnBPSe had decomposed. The TnBPSe decomposition reaction for the conversion of TnBPSe to TnBPO was further studied at various time points after hot injection by $^{31}$P NMR (Fig. 1). The percentage composition of TnBPO after 15 min was calculated to be 30% on average, which is consistent with the reaction yields of CdSe QDs using TnBPSe[16]. Surprisingly, we were unable to observe any other molecular form of selenium in solution by $^{77}$Se or $^{31}$P NMR (Supplementary Figs. 2 and 3). Previous mechanistic studies attribute the lack of a Se signal in NMR to formation of CdSe QDs, which would be NMR silent[5, 14, 17, 19, 27]. However, in the absence of Cd this explanation is not possible.

The presence of an insoluble red powder around the edges of reaction vials, similar to the color of selenium oxide, suggested that a gaseous selenium product resulting from the TnBPSe decomposition had been formed, which we hypothesized to be hydrogen selenide (H$_2$Se). Figure 2a shows a diagram of the experimental apparatus used to test for H$_2$Se. The thermal decomposition reaction of TnBPSe takes place in one flask, and any gases formed flow through a cannula to a second flask where they collected. For this experiment, TnBPSe was rapidly injected into hot octanoic acid in tetradecane (250 °C) in the first flask, exactly mimicking the reaction conditions for QD synthesis with the exclusion of cadmium. In the second collection flask, the cannula was submerged in carbon disulfide (CS$_2$) cooled in an ice bath, allowing the gaseous products to dissolve. H$_2$Se is soluble in carbon disulfide, and was collected for analysis by $^{77}$Se NMR spectroscopy. A positive control experiment was conducted where H$_2$Se was generated in the first flask by the well-known reaction of zinc selenide with hydrochloric acid[30], and collected in CS$_2$ in the second flask.

Comparison of the $^{77}$Se NMR spectra of the control experiment and the TnBPSe decomposition experiment, seen in Supplementary Figs. 4 and 5, show that both reactions contain matching peaks, consistent with the expected NMR peak shift for H$_2$Se in CS$_2$ (Supplementary Table 1)[30, 31]. The observation of Se

in the rightmost flask provides conclusive evidence that H$_2$Se is the selenium containing decomposition product of TnBPSe. Furthermore, we confirmed that the H$_2$Se produced from the TnBPSe decomposition can react to form CdSe nanoparticles, by using the cannula apparatus in Fig. 2a with cadmium oleate in tetradecane at 250 °C in the second flask replacing CS$_2$. H$_2$Se from the TnBPSe decomposition reaction in the first flask was bubbled through the cadmium oleate resulting in the synthesis of CdSe nanoparticles shown in Fig. 2c.

To quantify the amount of H$_2$Se produced, a coaxial insert NMR tube was employed to allow the use of an external $^{77}$Se NMR standard. Triphenylphosphine selenide (TPPSe) was chosen as the external standard because its chemical shift appears near to, but distinct from, that of H$_2$Se. The $^{77}$Se NMR showing the H$_2$Se singlet and TPPSe standard doublet can be seen in Fig. 2b. From $^{77}$Se NMR integrated peaks, we calculated an average of 36% conversion of TnBPSe to H$_2$Se after 15 min of reaction. The discrepancy between the conversion yields from $^{77}$Se NMR (36%) and $^{31}$P NMR (30%) is within experimental error (Supplementary Note 1). Interestingly, we also observed that hot injection of the phosphine-chalcogenide (as opposed to heating the phosphine-chalcogenide already in solution) doubles the conversion rate of TnBPSe, as shown in Supplementary Fig. 8. This finding indicates that the hot injection method, which is known to be important for a narrow particle size distribution[21], is directly contributing to the conversion kinetics of phosphine-chalcogenide precursors to H$_2$Se. Finally, we also observe that increasing the amount of excess carboxylic acid in solution increases the conversion of TnBPSe to TnBPO (Supplementary Fig. 9). This result indicates that carboxylic acid actively participates in the P=Se bond cleavage as has been proposed previously[14].

We suggest that tertiary phosphine decomposition is a critical and necessary first step in the formation of CdSe QDs. A proposed overall reaction scheme for this process is shown in Fig. 3a, with details of the progression of the reaction mechanism shown in Fig. 3b and arrow pushing in Supplementary Fig. 10. In this mechanism, carboxylic acid, present in excess in standard QD reactions, attacks the tertiary phosphine selenide, breaking the phosphine selenide double bond, allowing it to be protonated forming an selenophosphanyl carboxylate intermediate. The resulting carboxylate is activated by a second carboxylic acid leading to the formation of an oxonium ion. Subsequently, a tetrahedral intermediate is formed by nucleophilic attack of the

**a** Overall Reaction Scheme

**b** Step 1: Selenophophanyl carboxylate formation

Step 2: Oxonium ion formation

Step 3: Nucleophillic attack, formation of a tetrahedral intermediate

Step 4: Collapse of the tetrahedral intermediate, H$_2$Se release

**Fig. 3** Reaction scheme for H$_2$Se release. **a** The proposed reaction scheme for the decomposition of tertiary phosphine selenide in the absence of Cd. **b** The steps of the proposed reaction mechanism with the tertiary phosphine in green and carboxylic acids in blue and purple. Newly formed bonds are in black

**a**

**b**

**Fig. 4** Cadmium carboxylate reaction schemes. **a** The formation of cadmium carboxylate. **b** The thermal decarboxylation as proposed in this work

second carboxylate on the oxonium ion. The acidic proton of the oxonium ion is in close proximity to the selenium and is able to protonate it a second time, breaking the final P–Se bond and releasing H$_2$Se. The tetrahedral intermediate collapses to yield anhydride and tertiary phosphine oxide, known by-products of QD synthesis[14, 27, 28]. Chen et al.[22] observed the decomposition of phosphine sulfides to phosphine oxides in acidic conditions at temperatures greater than 150 °C, resulting in the release of hydrogen chalcogenide, showing that this mechanism is not unprecedented or implausible. In addition, the reaction was run in tetradecane, a non-coordinating solvent, ruling out a previously reported pathway where elemental Se in octadecene (ODE) reacts with the double bond in ODE to generate H$_2$Se at high temperatures[32, 33]. The use of octanoic acid eliminates the possibility of this alternative mechanism occurring since, unlike oleic acid, it has no double bonds. The reactivity of octanoic acid and oleic acid was found to be sufficiently similar (Supplementary

Fig. 11). Also, experiments were performed to check for the effect of hydrocarbon chain length and purity of reagents on the reactivity of the phosphine selenide. As shown in Supplementary Fig. 12, TOPSe had similar reaction kinetics to TnBPSe, indicating that within this range there was no effect. The purity of reagents did not significantly affect the rate of TXPSe decomposition, Supplementary Figs. 13–15.

Interestingly, when different carboxylates are used in the reaction: oleic acid and cadmium acetate, we observe a mixed anhydride species through GCMS (gas chromatography mass spectrometry, Supplementary Fig. 16). Mixed anhydrides that are formed through the reaction of acetic anhydride and oleic acid have a higher reactivity compared to symmetric anhydrides[34], and are in a fast equilibrium with symmetric anhydrides and free carboxylic acid (reaching equilibrium in <10 min at 100 °C)[35]. The presence of mixed anhydride species provides strong evidence that the anhydrides and free carboxylic acids are

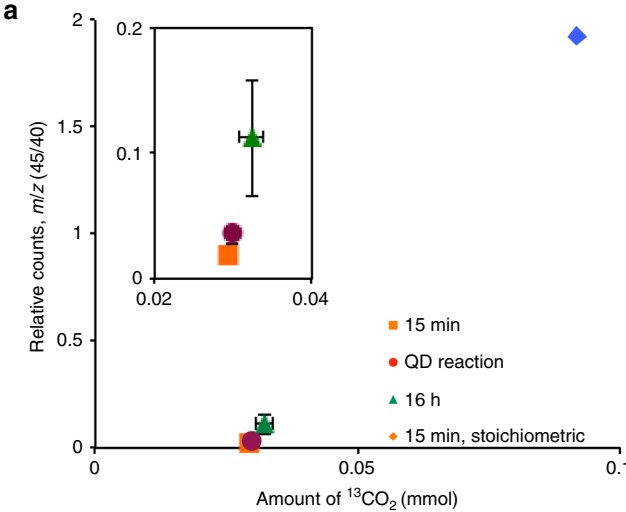

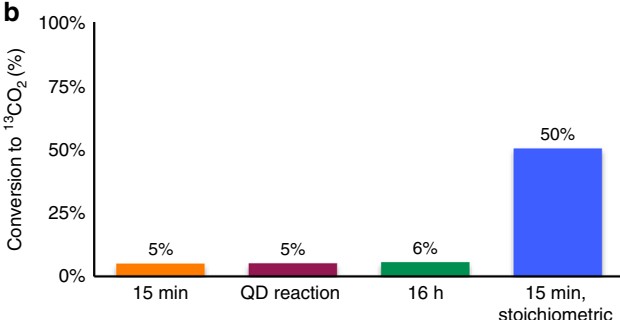

**Fig. 5** Detecting $^{13}CO_2$ from cadmium octanoate decomposition. **a** Headspace samples from different reaction conditions and the amount of $^{13}CO_2$ detected, the calibration curve to which these values are fit in Supplementary Fig. 20. **b** Bar graph of calculated percent conversion of cadmium carboxylate to $^{13}CO_2$ during the decomposition reaction. Samples compared include the headspace from; a Se-free decomposition with 12 times excess carboxylic acid after 15 min, a QD reaction, a Se-free decomposition with stoichiometric amounts of carboxylic acid after 15 min (QD absorbance spectrum in Supplemental Fig. 21, and a Se-free decomposition with excess carboxylic acid after 16 h. Tabular data can be found in Supplemental Table 2. Error bars depict the standard error of the mean of five replicate experiments

available to react during normal QD syntheses, perpetuating the mechanism described above.

**Thermal decomposition of cadmium octanoate**. In addition to the phosphine-chalcogenide, at elevated temperatures metal carboxylates can decompose by losing a carboxyl group and liberating carbon dioxide $(CO_2)$[36], leading us to consider the decomposition of the metal carboxylate as another possible reaction pathway en route to QD nucleation. Decomposition of cadmium carboxylate would form a highly reactive alkyl cadmium product (Fig. 4b), a historically common cadmium precursor for QD synthesis with tertiary phosphine selenides[2]. To test our hypothesis, the decomposition of cadmium octanoate under standard QD synthesis conditions was observed as a model for the decomposition of the common QD precursor, cadmium oleate. Alkyl cadmium is air sensitive, pyrophoric, very reactive, and not easily observed directly, so to verify the thermal decomposition of cadmium octanoate (Fig. 4b), we tracked the formation of $CO_2$.

As shown in Supplementary Fig. 17, $^{13}C$-labeled cadmium octanoate was used to discern $CO_2$ generated by the decarboxylation reaction from that of the atmosphere. GCMS analysis, shown in Fig. 5, revealed that $^{13}CO_2$ was present in the headspace of the reaction, indicating that the $^{13}C$-labeled cadmium octanoate had undergone decarboxylation, releasing $^{13}CO_2$ and forming alkyl cadmium (Fig. 4b). A description of the reaction vessel is present in Supplementary Fig. 18. Note that formation of $CO_2$ from metal carboxylates has also been postulated to arise from an alternative route that involves formation of metal carbonates or metal oxides (Supplementary Fig. 19)[28]. However, we observed no precipitation of cadmium oxide or cadmium carbonate in our reactions, signifying that the observed $CO_2$ was likely accompanied by formation of the metal alkyl.

GCMS was further used to quantify the amount of $^{13}CO_2$ that formed during reaction. We compared the amount of $^{13}CO_2$ in the headspace to a standard calibration curve made with varying amounts of $^{13}CO_2$, described in Supplementary Fig. 20. Somewhat surprisingly, the amount of $^{13}CO_2$ measured reached only 5% of that expected from total decarboxylation of all cadmium carboxylate in solution after 15 min. Even after 16 h the amount detected did not change appreciably, shown in Fig. 5b. Interestingly, when there is no excess carboxylic acid in solution, when the cadmium carboxylate is prepared using stoichiometric amounts of carboxylic acid to cadmium, the amount of $^{13}CO_2$ observed increases markedly to 50% (Fig. 5b).

This marked increase in reactivity suggests that the carboxylate is acting as a passivating agent, preventing the decomposition reaction to form the alkyl cadmium species, causing only small amounts of $CO_2$ to be released into the headspace. Rapid exchange of carboxylates at the metal center and QD surface are known[37], and equilibrium effects are a known limitation to thermal decarboxylation, along with thermal instability of the resulting product, or a preference for an alternative decomposition pathway[36]. Nonetheless, given that typical CdSe QD synthesis conditions are run in a large excess of carboxylate, it is unlikely that metal-carboxylate decomposition with subsequent metal-alkyl formation can account for the majority of the QDs formed.

## Discussion

It is interesting to relate our findings with previous studies of CdSe QD reaction mechanism. Here we have demonstrated that under the high reaction temperatures of typical QD syntheses, phosphine selenide and metal carboxylate decomposition produce highly reactive species that subsequently form QDs. The fact that QDs are formed in one flask containing metal carboxylate, with phosphine selenide precursors present in another flask, with just a cannula connecting the two flasks, indicates that phosphine selenide activation by a cadmium complex is not necessary for phosphine-chalcogenide bond cleavage. Although most previously proposed mechanisms have proposed an intermediate where both cadmium and selenium precursors remain intact and form the first initial CdSe bond[14, 16, 17, 19, 27, 38], our proposed mechanism first requires thermal decomposition of the Cd and Se molecular precursors before forming the first CdSe bond. Nonetheless, it is important to note that our study does not preclude the possibility that such direct interaction does occur and could also lead to the formation of QDs.

In a typical CdSe QD synthesis excess carboxylic acid causes rapid, non-uniform QD growth while inhibiting nucleation, resulting in fewer QDs of larger size[39–41]. This suggests an increased reactivity or availability of precursors during the growth

phase but inhibition of nucleation. However, theoretical models of nucleation kinetics predict no dependence on carboxylic acid concentration on nuclei formation from monomer[16, 42, 43]. These models assume monomer formation is first order in precursor concentration, not accounting for precursor reactions before monomer formation[42, 43]. Precursor conversion is an important step in QD reaction kinetics, as demonstrated by Hendricks and co-workers through their design of sulfur precursors with tune-able conversion kinetics[11, 16], dictating QD size and mono-dispersity. In this work, we found that increased carboxylic acid concentration resulted in increased tertiary phosphine selenide decomposition yielding more of the active precursor $H_2Se$. We also observed that excess carboxylic acid results in a decreased conversion of cadmium carboxylate to $CO_2$, indicating that decarboxylation is inhibited at high carboxylate concentrations. The converse dependencies on the concentration of carboxylic acid in the two precursor conversion reactions reported here suggest that there is a subtle balance between the two processes and each affect the nucleation and growth phases differently.

It is interesting to question how the rate of conversion of phosphine selenide is affected by the presence of Cd oleate in solution. The presence of Cd oleate could certainly change the kinetics of the phosphine decomposition, or act as a Lewis acid, as others have proposed[14, 15, 32], forming a Lewis acid-base complex as the first step to CdSe QD formation. To parse out the role of Cd as a Lewis acid we monitored the conversion of phosphine selenide to phosphine oxide in the presence of three equivalents (vs. TnBPSe) of zinc-, cadmium-, or silver-oleate. If the release of selenium was dependent on cadmium acting as a Lewis acid, we may expect silver (also a "soft" metal) to react similarly, but would expect zinc to be the least reactive because it is a harder metal. As seen in Supplementary Fig. 22, zinc follows the reactivity of the metal-free decomposition after 10 min. The lack of reactivity before 10 min is mirrored in ZnSe synthesis with Zn stearate reported by Li et al.[44] Silver oleate also had overall similar conversion to the metal-free case. However, with cadmium present we see a rapid increase in phosphine selenide conversion rate in the first minute after hot injection, Supplementary Fig. 23.

We attribute the initial fast consumption of phosphine selenide in the presence of cadmium to the burst nucleation of CdSe, which provides a large driving force for the release of selenium. Such an attribution has precedent, as it has been shown that the Cd and Se precursor reaction rates depend on the rate of QD nucleation[14,16,45]. According to our results, in the absence of cadmium $H_2Se$ gas is evolved. However, in the presence of a cadmium source $H_2Se$ rapidly reacts to form CdSe nuclei that precipitate from solution into a colloidal suspension. When this occurs as a burst nucleation event during hot injection, the removal of CdSe QDs as products in this reaction creates a large driving force for the precursor conversion reaction. This interpretation is consistent with the slowed precursor reaction kinetics observed for a heat up method of synthesis (Supplementary Fig. 8), as well as the similar kinetics observed for Cd after the first minute of hot injection (Supplementary Fig. 23).

The formation of the highly reactive species $H_2Se$ and a metal alkyl in situ during QD syntheses suggests that modern methods used to synthesize QDs are not actually that different from methods used decades ago. For example, some of the earliest reported syntheses of metal chalcogenide QDs involved injecting $H_2Se$ into a solution containing metal salts[46, 47]. And, in an early seminal breakthrough alkyl cadmium precursors with tertiary phosphine selenide enabled the synthesis of high-quality CdSe QDs with well-controlled size[2]. Our results indicate that QD syntheses have evolved such that these highly reactive molecular precursors used decades ago are likely still the reactive species, but are simply produced in a more controlled manner today. Rational design of metal and chalcogen precursors that decompose into metal alkyl or hydrogen selenide species in a controlled manner could be a possible route to significant future breakthroughs in QD synthesis.

In summary, we have determined that no less than three separate mechanisms contribute to the formation of metal chalcogenide bonds during the formation of CdSe and related QDs. Analysis of independent thermal decomposition reactions of cadmium carboxylate and tertiary phosphine selenide suggests that each organometallic precursor decomposes independent of one another in standard QD synthesis conditions to form highly reactive species, alkyl cadmium and hydrogen selenide, respectively. A third mechanism does not involve precursor decomposition and would include secondary phosphine-chalcogenide impurities reacting with metal carboxylates[5]. The decomposition of tertiary phosphine-chalcogenide to form tertiary phosphine oxide and $H_2Se$ at a high yield makes this pathway likely the dominant one. The alkyl cadmium decomposition pathway and the secondary phosphine-chalcogenide impurity pathway are likely much less important but can account for upwards of 10% of the synthesized QDs. These findings are important in that they provide a rational roadmap for improving the current approach to designing better QD synthetic methods. By understanding the chemistry that leads to the initial QD nucleation event, new organometallic precursors can be designed to control this decomposition to yield highly engineered QDs of desired sizes, shapes, properties, and chemical compositions.

## Methods

**Synthesis of selenium precursor.** Tri-*n*-butylphosphine selenide was prepared by mixing selenium pellets with tri-*n*-butylphosphine (99%) in toluene under a nitrogen atmosphere to make a 1 M solution, heating at 60 °C until homogeneous.

**Tertiary phosphine selenide decomposition.** Tri-*n*-butylphosphine selenide (2 mmol) was injected into a flask of octanoic acid (4 mmol) in tetradecane (8 mL) at 250 °C under a nitrogen atmosphere for 15 min. The resulting products in solution were analyzed by $^1H$, $^{31}P$, and $^{77}Se$ NMR at different time points and carboxylic acid concentrations. A cannula transfer apparatus was assembled as in Fig. 2a so that the gases from the TnBPSe decomposition flask were bubbled through the solution in the second flask. The gaseous products were collected in carbon disulfide (12 mL, $CS_2$) at 0 °C (ice bath) in the second flask. The solution reacted for 15 min, the amount of time necessary to make CdSe QDs in the same apparatus. The solution in $CS_2$ was then prepared in a NMR tube with a STEM coaxial insert and a triphenylphosphine selenide (TPPSe) external standard. A control experiment was conducted in parallel to allow for comparison of spectra; a $CS_2$ solution of hydrogen selenide was generated by reaction of zinc selenide with a dropwise addition hydrochloric acid in the first flask, as reported by Schneider and Wieghardt.[30] In addition, the second flask was replaced with Cd oleate in tetradecane and resulted in the formation of CdSe nanocrystals.

**Cadmium octanoate decomposition.** $^{13}C$-labeled cadmium octanoate was prepared by heating 1-$^{13}C$-octanoic acid (4 mmol) and cadmium oxide (2 mmol) in tetradecane by heating to 210 °C for 30 min in a schlenk flask. The resulting clear, colorless liquid was degassed and the flask was sealed. The side arm of the flask was sealed with a septum and purged with nitrogen. The sealed flask was then heated to 280 °C for 15 min, analogous to the conditions of a QD synthesis. The headspace of the reaction was analyzed using GCMS. The purged space in the arm of the flask was analyzed to ascertain a background concentration of gases. The Teflon pin of the flask was then opened allowing the headspace of the reaction to mix with the atmosphere in the arm of the flask. A second sample of gas was taken through the septum and the relative concentrations of gases before and after mixing was determined. A calibration curve was created by varying the amounts of $^{13}CO_2$ added to a schlenk flask filled with an equivalent volume of tetradecane as in the experimental unknowns with an argon internal standard was added equally across all samples. The ratio of counts vs. the internal standard was calculated by dividing the counts of $^{13}CO_2$ ($m/z = 45$) by those of argon ($m/z = 40$). The average for each concentration was taken over five replicates.

**NMR characterization.** Proton ($^1H$), carbon ($^{13}C$), phosphorus ($^{31}P$), and selenium ($^{77}Se$) NMR spectra were recorded at ambient temperature on an Avance 500

(500 MHz) spectrometer. Chemical shift ($\delta$) is recorded in ppm and coupling constants ($J$) are reported in Hertz (Hz).

**GCMS characterization**. Headspace samples were measured on a Shimadzu GCMS–QP2010. The instrument was operated with an injection temperature of 250 °C, a column temperature ramping from 40–240 °C, an ion source temperature of 225 °C, and a column flow rate of 17 psi helium carrier gas at 1.05 mL/min.

**Data availability**. The data that support the findings of this study are available from the corresponding author upon reasonable request.

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

## Acknowledgements

We acknowledge funding form the NSF (CHE-1609365). Jalil Shojaie and Terry O'Connell (Univ. of Rochester) are thanked for their assistance in conducting the GCMS experiments. We thank Chris Evans and Brett Swartz for preliminary studies in this research direction. We also thank William B. Jones, Robert K. Boeckman Jr., and Nicholas J. Gower (Univ. of Rochester) for helpful discussions.

## Author contributions

L.C.F. and T.D.K. conceived of the experiments, analyzed the results, and wrote the paper. L.C.F. conducted the experiments.

## Additional information

**Competing interests:** The authors declare no competing financial interests.

