## [Peer Review File · Nature Communications]

Reviewers' comments:

Reviewer #1 (Remarks to the Author):

Authors showed that typical precursors used in CdSe synthesis each decompose into highly reactive H₂Se and alkyl cadmium. These results contribute to a better understanding of CdSe formation and should allow for better control of CdSe monomer formation. The science used in this paper is sound with many control experiments performed to exclude other possibilities. Given the wide applicability and scientific merit of this study, I recommend publication.

The authors used shorter chain analogues of the more commonly used TOP and oleic acid. Is there an effect when longer chain reagents are used such as different reaction kinetics or conversion percentages? Oleic acid has a double bond that should also react with selenium to produce H₂Se, so a comparison would be interesting.

Resolution of all NMR in the main text is very poor such that it is difficult to discern the trends. The NMR in the supporting info have good resolution. Figure 3b should be "2 CO₂".

Reviewer #2 (Remarks to the Author):

I am always excited when nanocrystal chemists take the time to really understand the details of the chemical transformations that occur in syntheses. Frankly, we've been sloppy chemists for too long, and the excellent effort by the authors has potential to be truly impactful on 1. how we think about these reactions and how to control them 2. Improving safety in preparations (especially in larger scale-up reactions) 3. And improving the reproducibility in reactions. It also sets a bar for the type of studies that should be done more often by the community. While I don't think the manuscript is of Nature Communications calibre yet, there are some experiments suggested below that could take it there.

Abstract:

"chemical precursors that were used in the most common synthetic methods do not directly react to form QD monomers..." This is too strong a statement. The authors have not shown this, but rather that decomposition products of these monomers definitely form QDs and are important. Who is to say the chemical precursors don't also directly react?

Pg 4. "Surprisingly we were unable to observe any other molecular form of selenium in solution by NMR." Presumably ⁷⁷Se NMR? Please clarify language. Should there be a ⁷⁷Se NMR to go with Figure S2, i.e., of the original solution after decomposition?

Figure 1: This figure shows that the formation of TnBPO correlates with the loss of TnBPSe. This experiment was done in a "control" solution without the presence of Cd. Coming back to my comment about the abstract, the authors have shown these decompositions are important in control experiments, but haven't really shown if these mechanistic routes are the only or the most active routes in nanocrystal formation. I would be much more convinced that H₂Se is the MOST active route in real nanocrystal syntheses if the kinetic measurements shown in Figure 1 were duplicated in the presence of Cd. Should these traces lie on top of one another, then we could be more confident that H₂Se is the dominant active Se reagent in nanocrystal formation and that Cd is not intimately involved in a secondary route in the TnBPSe decomposition.

I have some concerns about changing the acid from oleic acid to a "cleaner" fatty acid. The technical grade oleic acid used here is "dirty" (<90%). I would like to see evidence that the common impurities of Oleic acid are not important to nanocrystal formation. A carefully designed experiment would be worth it to splurge on some >99% OA.

I am also concerned about exchanging an unsaturated fatty acid (oleic acid) for a saturated one. The authors mention how the double bond of Octadecene has been shown to be a reducing agent for Se to form H₂Se (pg 7, near the bottom). It is possible that the double bond of oleic acid may also do so. Experiments should be performed to show that this route is not active under typical CdSe formation conditions.

Page 9, Figure 3b. Are we sure that Dicarboxocadmium(II) goes all the way to dialkylcadmium(II)? How important is the mixed intermediate alkylcarboxocadmium(II) species? Simple ¹H NMR of the solution could tell you a lot about this dynamic, especially since CO₂ formation seems so substoichiometric.

Page 9 "we observed no evidence for these insoluble compounds" Is the evidence that you did not see a precipitate? The conditions of the reactions are designed to make colloiddally stable particles of CdSe, which is also an insoluble compound! Please clarify.

Page 11, first paragraph. There is some funny logic here that seems to break Le Chatelier's principle. Why would alkyl cadmium react with carboxylic acid and CO₂ to remake cadmium carboxylate? What is the carboxylic acid's job here?

I spent a few minutes with March's Advanced Organic Chemistry and while I was not able to immediately come up with an alternative explanation, there is a lot of chemical history out there on this topic. I think the authors could benefit from digging through some classic organic chemistry to learn mechanistic details of how decarboxylation happens on metals such as Cu or in the presence of base.

Figure 1 caption: wrong "its"

Page 6 appears near to, but distinct from,... (I think commas are needed)

Page 7, p2, In 2 Figure 2a does not show a mechanism but rather a balanced equation.

As well, it seems strange to have this lengthy discussion of mechanism here with the figure hidden in the SI. Can these two be brought together either in the main text or the SI?

Page 9. Figure 3a. Balance the 1st equation with water. Balance the second with 2CO₂

Page 11, first sentence: awkward. Try and reword.

In general, there are quite a few missing commas and some awkward sentences, etc. Please have a fresh pair of eyes read the manuscript over critically.

DEPARTMENT OF CHEMISTRY

Todd D. Krauss

Professor of Chemistry and Chair

Professor of Optics

We thank the reviewers for their careful review of our work and for providing constructive feedback. We are pleased to see that both reviewers found our work to be interesting and impactful and of high scholarly merit. The reviewers suggested some additional experiments to further demonstrate the importance of our proposed mechanism and recommended revisions to the figures and the manuscript. We have revised the manuscript to address their suggestions as described below.

Reviewer #1:

Authors showed that typical precursors used in CdSe synthesis each decompose into highly reactive H₂Se and alkyl cadmium. These results contribute to a better understanding of CdSe formation and should allow for better control of CdSe monomer formation. The science used in this paper is sound with many control experiments performed to exclude other possibilities. Given the wide applicability and scientific merit of this study, I recommend publication.

1. The authors used shorter chain analogues of the more commonly used TOP and oleic acid. Is there an effect when longer chain reagents are used such as different reaction kinetics or conversion percentages? Oleic acid has a double bond that should also react with selenium to produce H₂Se, so a comparison would be interesting.

We thank the reviewer for this suggestion; we also find the comparison interesting. We originally used shorter chain analogues for both the tertiary phosphine and carboxylic acid to eliminate additional reaction pathways that could occur through reaction with the double bond or other impurities in the commonly used technical grade (90%) oleic acid. We recognize that a comparison to the more commonly used reagents strengthens our argument that these are indeed the mechanisms taking place and is an important addition to the manuscript.

A comparison of the TnBPSe and TOPSe under identical reaction conditions showed only a slight increase in activity with the shorter chain tertiary phosphine, indicating that these reactions are proceeding by the same reaction pathways. It is likely that steric hindrance is the cause for the decreased rate in the longer chain case. A peak in the ³¹P NMR at 49 ppm was observed in the reactions run with

University of Rochester
Rochester, NY 14627-0216

Hutchison Hall 465 (585) 275-5093
FAX: (585) 276-0205
krauss@chem.rochester.edu

TOPSe (90%) but not those in TnBPSe (99%). This is an impurity but the intensity of the peak does not change throughout the reaction (3%) indicating that the species is not participating in the reaction.

Figure S12. A NMR analysis following decomposition of TnBPSe after hot injection, the percent composition of TXPSe (top) and TXPO (bottom) in the CdSe QD forming reaction over time, comparing the reactivity of TOPSe and TnBPSe. Trend lines are provided as guides to the eye.

Figure S13. ^{31}P NMR spectrum of the TOPSe reaction mixture over time, showing the amount of TOPSe (37 ppm) decreases as the amount of TOPO (55 ppm) increases. Also shown is an impurity peak at 49 ppm that does not change integration over time and is approximately 3% of the TOPSe solution. This peak is not observed with higher purity phosphines.

Additionally, 97% TnBPSe was used in the original manuscript but for these revision experiments 99% TnBPSe was used. We observed no difference in reactivity between the two grades of reagent.

Figure S14. A NMR analysis following decomposition of TnBPSe after hot injection, the percent composition of TXPSe (top) and TXPO (bottom) in the decomposition reaction over time, comparing the reactivity of 97% and 99% TnBPSe. Trend lines are provided as guides to the eye.

We compared the rate of phosphine decomposition in the presence of equivalent amounts of Octanoic acid (99%), Oleic Acid (90%) and Stearic Acid (95%) and found that the differences in reactivity were minimal and within experimental error. As a general trend, Oleic acid is the slowest to react of the three, indicating that the double bond is not aiding in the release of H₂Se in this case, but perhaps slowing down our proposed pathway via interaction with the double bond or steric bulk. We used stearic acid as a saturated analogue to oleic acid. Initially we observed a rate similar to that of octanoic acid (also saturated). However, stearic acid is a waxy solid at room temperature and samples at later time points had to be reheated to prepare the samples for NMR, which unfortunately changed the conversion rate for these samples and thus we could not include them in the plot. Overall we saw no appreciable difference in reactivity upon changing carboxylic acid.

Figure S11. A NMR analysis following decomposition of TnBPSe after hot injection. The percent composition of TXPSe (top) and TXPO (bottom) in the decomposition reaction over time in the presence of excess Oleic, Stearic, or Octanoic Acid. Trend lines are provided as guides to the eye.

The following changes have been made to the manuscript:

1) These figures have been added to the supplementary information.

2) On page 7 we added the following:

“Additionally, the reaction was run in tetradecane, a non-coordinating solvent, ruling out a previously reported pathway where elemental Se in octadecene (ODE) reacts with the double bond in ODE to generate H₂Se at high temperatures.^{36,37} The use of octanoic acid eliminates the possibility of this alternative mechanism occurring since, unlike oleic acid, it has no double bonds. The reactivity of octanoic acid and oleic acid was found to be sufficiently similar (Supplementary Figure S11). Also, experiments were performed to check for the effect of hydrocarbon chain length on the reactivity of the phosphine selenide. As shown in Supplementary Figure S12, TOPSe had similar reaction kinetics to TnBPSe, indicating that within this range there was no effect.”

2. Resolution of all NMR in the main text is very poor such that it is difficult to discern the trends. The NMR in the supporting info have good resolution.

Figures 1 and 2 were updated to include high resolution NMR data for clarity. We thank the reviewer for catching this oversight.

3. Figure 3b should be “2 CO₂”.

Figure 3 was altered to include all products of the balanced reactions. We thank the reviewer for catching our error.

Reviewer #2 (Remarks to the Author):

I am always excited when nanocrystal chemists take the time to really understand the details of the chemical transformations that occur in syntheses. Frankly, we've been sloppy chemists for too long, and the excellent effort by the authors has potential to be truly impactful on 1. how we think about these reactions and how to control them 2. Improving safety in preparations (especially in larger scale-up reactions) 3. And improving the reproducibility in reactions. It also sets a bar for the type of studies that should be done more often by the community. While I don't think the manuscript is of Nature Communications calibre yet, there are some experiments suggested below that could take it there.

1. Abstract:

"chemical precursors that were used in the most common synthetic methods do not directly react to form QD monomers..." This is too strong a statement. The authors have not shown this, but rather that decomposition products of these monomers definitely form QDs and are important. Who is to say the chemical precursors don't also directly react?

The reviewer has a good point and we agree completely. We altered the strength of this statement and hope this change in language presents a more accurate description of our findings.

On page 1 (abstract) we added the following:

"Remarkably, we find that it is not necessary for the chemical precursors used in the most common synthetic methods to directly react to form QD monomers, but rather they can generate in situ the same highly reactive Cd and Se precursors that were used in some of the original II-VI QD syntheses decades ago, i.e. hydrogen chalcogenide gas and alkyl cadmium."

2. Pg. 4. "Surprisingly we were unable to observe any other molecular for of selenium in solution by NMR." Presumably ^{77}Se NMR? Please clarify language. Should there be a ^{77}Se NMR to go with Figure S2, i.e., of the original solution after decomposition?

We thank the reviewer for identifying this oversight. We've clarified in the text (page 4) that we indeed meant ^{77}Se NMR, and have added the ^{77}Se NMR (Supplementary Figure S3) of TnBPSe before and after the decomposition reaction to the SI. This compliments the ^{31}P NMR in Figure S2 and shows a complete picture of the phosphorous and selenium species in solution. As you can see in Figure S3 (below) TnBPSe is the only selenium species in solution before and after the decomposition reaction.

^{77}Se NMR

Figure S3. ^{77}Se NMR of TnBPSe in toluene-*d*8 before and after 15 minutes of reaction at 250°C, remaining in Flask 1 of the cannula transfer apparatus. Chemical shifts are reported in Table S1.

3. Figure 1: This figure shows that the formation of TnBPO correlates with the loss of TnBPSe. This experiment was done in a “control” solution without the presence of Cd. Coming back to my comment about the abstract, the authors have shown these decompositions are important in control experiments, but haven’t really shown if these mechanistic routes are the only or the most active routes in nanocrystal formation. I would be much more convinced that H₂Se is the MOST active route in real nanocrystal syntheses if the kinetic measurements shown in Figure 1 were duplicated in the presence of Cd. Should these traces lie on top of one another, then we could be more confident that H₂Se is the dominant active Se reagent in nanocrystal formation and that that Cd is not intimately involved in a secondary route in the TnBPSe decomposition.

The reviewer is rightly asking about the relative importance of the phosphine selenide decomposition under actual CdSe QD growth conditions, i.e. with Cd Oleate present in the flask. This of course is a challenging problem because the presence of Cd Oleate certainly could change the kinetics of the phosphine decomposition. Interestingly, when we added Cd Oleate to the phosphine decomposition reaction, we found after nucleation the phosphine selenide reaction rate proceeds at a similar rate to the decomposition reaction in the absence of cadmium (Figure S23). However, with Cd present we see a rapid increase in rate in the first minute after hot injection.

One hint to understanding what may be occurring in the presence of Cd comes from the fact that we observed the phosphine selenide reaction proceeds much slower using a heat up method (where reactants are added and heated), rather than hot injection (where the phosphine selenide is injected into a hot solvent) (Figure S8 and Ref. 33). The heat up method being slower suggests that burst nucleation of CdSe, which happens during hot injection, likely plays a key role in the reaction kinetics of the phosphine. In this context, it is not surprising that the initial reaction rate would increase in the presence of cadmium. The nucleation of CdSe provides a large driving force for the release of selenium that is much larger than the evolution of H₂Se gas.

Figure S23. A NMR analysis following decomposition of TnBPSe after hot injection. The percent composition of TXPSe (top) and TXPO (bottom) in the decomposition reaction over time, is shown comparing the reactivity with and without cadmium (1 eq) present. Trend lines are provided as guides to the eye.

Previous studies have attributed the conversion of the phosphine selenide to phosphine oxide during the production of CdSe QDs as being due to Cd oleate and TOPSe forming a Lewis Acid-Base complex [Ref. 14-16 in Main Text]. In an attempt to parse out the role of Cd as a Lewis acid we repeated the previous experiment in the presence of three equivalents (versus TnBPSe) of zinc oleate and Silver oleate and compared it to the same reaction conditions in the presence of three equivalents of cadmium oleate. The reaction conditions used were optimal for CdSe QD formation but not optimized for the silver or zinc reactions. Cadmium and silver are both “soft” metals while Zinc is classified as an “intermediate” metal. As seen in Figure S22, zinc follows the reactivity of the metal free decomposition after 10 minutes. The lack of reactivity before 10 minutes is mirrored in ZnSe synthesis with Zn stearate reported by Peng et al. (Ref. 48). The presence of silver oleate caused an initial increase in conversion at early times, but overall similar conversion was observed to the metal free case. Importantly, we saw no evidence of the formation of ZnSe nanoparticles (before 8 minutes) or AgSe₂ nanoparticles. Because of the lack of production of nanoparticles formed initially, we propose that rapid nucleation is not occurring upon hot injection of TnBPSe into silver or zinc oleate at 250°C, and thus the TnBPSe decomposition proceeds similarly to if there was no metal present. If the release of selenium was dependent on cadmium acting as a Lewis acid, we would expect silver to react similarly causing an increase in the conversion of TnBPSe to TnBPO. We would expect zinc oleate to be the least reactive because it is not as soft of a metal. Due to the similarly low reactivity of the silver and zinc oleate, we propose that the driving force

of nucleation has a much larger effect on the release of selenium than the coordination of cadmium to selenium as a Lewis acid.

Figure S22. A NMR analysis following conversion of TnBPSe to TnBPO after hot injection. The percent composition TXPO in the decomposition reaction over time is shown, comparing the reactivity with three equivalents of cadmium, silver, or zinc oleate to that of the reaction in the absence of metal.

On page 13 we added the following:

“It is interesting to question how the rate of conversion of phosphine selenide is affected by the presence of Cd oleate in solution. The presence of Cd oleate could certainly change the kinetics of the phosphine decomposition, or act as a Lewis acid, as others have proposed,^{14,15,32} forming a Lewis acid-base complex as the first step to CdSe QD formation. To parse out the role of Cd as a Lewis acid we monitored the conversion of phosphine selenide to phosphine oxide in the presence of three equivalents (versus TnBPSe) of zinc-, cadmium- or silver- oleate. If the release of selenium was dependent on cadmium acting as a Lewis acid, we may expect silver (also a “soft” metal) to react similarly, but would expect zinc to be the least reactive because it is a harder metal. As seen in Supplementary Figure S22, zinc follows the reactivity of the metal-free decomposition after 10 minutes. The lack of reactivity before 10 minutes is mirrored in ZnSe synthesis with Zn stearate reported by Li et al.⁴⁸ Silver oleate also had overall similar conversion to the metal-free case. However, with cadmium present we see a rapid increase in phosphine selenide conversion rate in the first minute after hot injection, Supplementary Figure S23.

We attribute the initial fast consumption of phosphine selenide in the presence of cadmium to the burst nucleation of CdSe, which provides a large driving force for the release of selenium. Such an attribution has precedent, as it has been shown that the Cd and Se precursor reaction rates depend on

the rate of QD nucleation.^{14,16,49} According to our results, in the absence of cadmium H_2Se gas is evolved. However, in the presence of a cadmium source H_2Se rapidly reacts to form CdSe nuclei that precipitate from solution into a colloidal suspension. When this occurs as a burst nucleation event during hot injection, the removal of CdSe QDs as products in this reaction creates a large driving force for the precursor conversion reaction. This interpretation is consistent with the slowed precursor reaction kinetics observed for a heat up method of synthesis (Supplementary Figure S8), as well as the similar kinetics observed for Cd after the first minute of hot injection (Supplementary Figure S23).“

4. I have some concerns about changing the acid from oleic acid to a “cleaner” fatty acid. The technical grade oleic acid used here is “dirty” (<90%). I would like to see evidence that the common impurities of Oleic acid are not important to nanocrystal formation. A carefully designed experiment would be worth it to splurge on some >99% OA.

I am also concerned about exchanging an unsaturated fatty acid (oleic acid) for a saturated one. The authors mention how the double bond of Octadecene has been shown to be a reducing agent for Se to form H_2Se (pg. 7, near the bottom). It is possible that the double bond of oleic acid may also do so. Experiments should be performed to show that this route is not active under typical CdSe formation conditions.

We agree that reagent impurities are important to consider in these QD reaction mechanisms, which is why we took steps to avoid them in our experimental design. The technical grade oleic acid used in the original manuscript was used to make Cadmium Oleate for the cannula experiment that resulted in CdSe QDs. Octanoic acid (99%) was used for all other experiments. Nevertheless, we agreed that this deviation from the typical precursors used in QD synthesis could affect the resulting reaction kinetics and have compared octanoic acid, oleic acid, and steric acid, discussed above in response to reviewer 1’s similar suggestion (Comment 1) and have included those results in the supplementary information and referenced them in the main text.

On page 7 we added the following:

“Additionally, the reaction was run in tetradecane, a non-coordinating solvent, ruling out a previously reported pathway where elemental Se in octadecene (ODE) reacts with the double bond in ODE to generate H_2Se at high temperatures.^{36,37} The use of octanoic acid eliminates the possibility of this alternative mechanism occurring since, unlike oleic acid, it has no double bonds. The reactivity of octanoic acid and oleic acid was found to be sufficiently similar (Supplementary Figure S11). Also, experiments were performed to check for the effect of hydrocarbon chain length on the reactivity of the phosphine selenide. As shown in Supplementary Figure S12, TOPSe had similar reaction kinetics to TnBPSe, indicating that within this range there was no effect.”

Additionally, we repeated the TnBPSe decomposition reaction with 99% Oleic acid and compared to 90% oleic acid to eliminate any possibility of impurity involvement. We observed that the 99% oleic acid increased the TnBPSe conversion rate, suggesting that impurities in the Oleic acid could slow the conversion reaction. This figure has been added to the supplementary information.

On page 8 we added the following:

“The purity of reagents did not significantly affect the rate of TXPSe decomposition, Supplementary Figures S13-15.”

Figure S15. A NMR analysis following decomposition of TnBPSe after hot injection. a.) The percent composition of TXPSe (top) and TXPO (bottom) in the decomposition reaction over time, comparing the reactivity of 90% and 99% Oleic Acid. Trend lines are provided as guides to the eye.

6. Page 9, Figure 3b. Are we sure that Dicarboxocadmium (II) goes all the way to dialkylcadmium (II)? How important is the mixed intermediate alkylcarboxocadmium (II) species? Simple ^1H NMR of the solution could tell you a lot about this dynamic, especially since CO_2 formation seems so substoichiometric.

We agree with the reviewer that experiments observing the dynamics of this reaction via ^1H NMR would be interesting and informative. However, the cadmium carboxylate solution is a waxy solid at room temperature precluding the ability to carry out these studies. We believe these dialkylcadmium species to be so transient and reactive that they would not be able to be observed through quenching at varied timepoints as was possible with the phosphine decomposition experiment.

7. Page 9 “we observed no evidence for these insoluble compounds” Is the evidence that you did not see a precipitate? The conditions of the reactions are designed to make colloiddally stable particles of CdSe, which is also an insoluble compound! Please clarify.

We did not observe cadmium oxide or cadmium carbonate as precipitates in solution. To test solubility we stirred a solution of each in Tetradecane at 250°C and found neither to be soluble under these conditions, the precipitate (white-Cd Carbonate or brown-Cd Oxide) remained in both experiments.

On page 10 we added the following:

“However, we observed no precipitation of cadmium oxide or cadmium carbonate in our reactions, signifying that the observed CO₂ was likely accompanied by formation of the metal alkyl.”

8. Page 11, first paragraph. There is some funny logic here that seems to break Le Chatelier's principle. Why would alkyl cadmium react with carboxylic acid and CO₂ to remake cadmium carboxylate? What is the carboxylic acid's job here?

I spent a few minutes with March's Advanced Organic Chemistry and while I was not able to immediately come up with an alternative explanation, there is a lot of chemical history out there on this topic. I think the authors could benefit from digging through some classic organic chemistry to learn mechanistic details of how decarboxylation happens on metals such as Cu or in the presence of base.

We agree that the equilibrium argument lacks strength and have altered the text to clarify what we were trying to convey, that the excess carboxylic acid is preventing the decomposition of cadmium carboxylate through exchange on the cadmium centre, passivating it from further reaction. We were also surprised by the lack of mechanistic information regarding decarboxylation of metal carboxylates, especially late transition metals like Cadmium. We did find supportive evidence of our proposed mechanism in looking at mercury analogues (Ref. 40 in the main text). In general we found many contradictory accounts of the decomposition process and reaction products in the literature: for example, a thermal decarboxylation pathway and the decomposition to cadmium oxide through the cadmium carbonate, which is a low temperature pathway. At the high temperatures required for QD syntheses, and with our evidence against the formation of cadmium oxide and cadmium carbonate, we are confident we are in the thermal decomposition regime. (Mehrotra, R. C.; Bohra, R. *Metal Carboxylates*; Academic Press: London, 1983.)

We modified the manuscript on page 11 (now 12) as follows

“This dramatic increase in reactivity suggests that the carboxylate is acting as a passivating agent, preventing the decomposition reaction to form the alkyl cadmium species, causing only small amounts of CO₂ to be released into the headspace. Rapid exchange of carboxylates at the metal centre and QD surface are known,⁴¹ and equilibrium effects are a known limitation to thermal decarboxylation, along with thermal instability of the resulting product, or a preference for an alternative decomposition pathway.⁴⁰ Nonetheless, given that typical CdSe QD synthesis conditions are run in a large excess of carboxylate, it is unlikely that metal-carboxylate decomposition with subsequent metal-alkyl formation can account for the majority of the QDs formed.”

9. As well, it seems strange to have this lengthy discussion of mechanism here with the figure hidden in the SI. Can these two be brought together either in the main text or the SI?

We thank the reviewer for this suggestion. We think the discussion of the mechanism is important and should be included in the main text but we feel that the arrow pushing mechanism figure (Figure S10) would not be a good fit, which is why we originally included the overall reaction scheme in Figure 2a. We have made an additional figure for the main text that follows the mechanism step by step including the intermediates we describe in the main text. We also removed the overall reaction scheme from 2a and included it in the new figure. See new Figure 3 below.

Overall Reaction Scheme

Step 1: Selenophosphanyl carboxylate formation

Step 2: Oxonium ion formation

Step 3: Nucleophilic attack, formation of a tetrahedral intermediate

Step 4: Collapse of the tetrahedral intermediate, H₂Se release

Figure 3. Top: The proposed reaction scheme for the decomposition of tertiary phosphine selenide Bottom: The steps of the proposed reaction mechanism with the tertiary phosphine in green and carboxylic acids in blue and purple. Newly formed bonds are in black

10. Page 9. Figure 3a. Balance the 1st equation with water. Balance the second with 2CO₂

Figure 3 (Now Figure 4) was altered to include all products of the balanced reactions. We thank the reviewer for catching this mistake.

11. Figure 1 caption: wrong "its"

We decided that the word did not add clarity to the sentence, so we deleted the word "its"

On page 4, figure 1 it now reads:

"A NMR analysis following decomposition of TnBPSe after hot injection"

Page 6 appears near to, but distinct from,... (I think commas are needed)

We agree and added commas appropriately on page 6:

“Triphenylphosphine selenide (TPPSe) was chosen as the external standard because its chemical shift appears near to, but distinct from, that of H₂Se.”

Page 7, p2, In 2 Figure 2a does not show a mechanism but rather a balanced equation

We altered figure 2 and added the mechanism in Figure 3, helping to address this comment and Reviewer 2's comment 9

Page 11, first sentence: awkward. Try and reword.

The first sentence on page 11 was changed in response to Reviewer 2's Comment 8 and now reads more clearly.

The first sentence on page 11 now reads:

“This dramatic increase in reactivity suggests that the carboxylate is acting as a passivating agent, preventing the decomposition reaction to form the alkyl cadmium species, causing only small amounts of CO₂ to be released into the headspace.”

In general, there are quite a few missing commas and some awkward sentences, etc. Please have a fresh pair of eyes read the manuscript over critically.

The suggested revisions were made in addition to other grammatical changes and changes in word choice for clarity. We thank the reviewer for catching these errors.

REVIEWERS' COMMENTS:

Reviewer #1 (Remarks to the Author):

The authors have addressed this reviewer's previous concerns and this manuscript is now in good shape for publication.

Reviewer #2 (Remarks to the Author):

I'm satisfied by the careful consideration and response to both the reviewer's comments. I can now recommend publication.